# Perspectives for Future Use of Extracellular Vesicles from Umbilical Cord- and Adipose Tissue-Derived Mesenchymal Stem/Stromal Cells in Regenerative Therapies—Synthetic Review

**DOI:** 10.3390/ijms21030799

**Published:** 2020-01-25

**Authors:** Joanna Lelek, Ewa K. Zuba-Surma

**Affiliations:** 1Laboratory of Stem Cell Biotechnology, Malopolska Centre of Biotechnology, Jagiellonian University, 30-387 Krakow, Poland; joanna.lelek@uj.edu.pl; 2Department of Cell Biology, Faculty of Biochemistry, Biophysics, and Biotechnology, Jagiellonian University, 30-387 Krakow, Poland

**Keywords:** extracellular vesicles, mesenchymal stem cells, adipose tissue cells, Wharton’s jelly, tissue repair

## Abstract

Mesenchymal stem/ stromal cells (MSCs) represent progenitor cells of various origin with multiple differentiation potential, representing the most studied population of stem cells in both in vivo pre-clinical and clinical studies. MSCs may be found in many tissue sources including extensively studied adipose tissue (ADSCs) and umbilical cord Wharton’s jelly (UC-MSCs). Most of sanative effects of MSCs are due to their paracrine activity, which includes also release of extracellular vesicles (EVs). EVs are small, round cellular derivatives carrying lipids, proteins, and nucleic acids including various classes of RNAs. Due to several advantages of EVs when compare to their parental cells, MSC-derived EVs are currently drawing attention of several laboratories as potential new tools in tissue repair. This review focuses on pro-regenerative properties of EVs derived from ADSCs and UC-MSCs. We provide a synthetic summary of research conducted in vitro and in vivo by employing animal models and within initial clinical trials focusing on neurological, cardiovascular, liver, kidney, and skin diseases. The summarized studies provide encouraging evidence about MSC-EVs pro-regenerative capacity in various models of diseases, mediated by several mechanisms. Although, direct molecular mechanisms of MSC-EV action are still under investigation, the current growing data strongly indicates their potential future usefulness for tissue repair.

## 1. Introduction 

Mesenchymal stem cells (MSCs), also known as mesenchymal stromal cells, are progenitor cells known since 1966 when have been described in human bone marrow [1]. Cells identified as MSCs have to meet the criteria specified by international scientific community such as (i) adherence to plastic, (ii) expression of specific surface antigens (including CD105, CD73, CD90), (iii) lack of hematopoietic antigens (e.g., for human: CD45, CD34, CD14, CD11b, human leukocyte antigens ((HLA)-DR), and (iv) exhibit potential to differentiate into multiple tissues—predominantly with mesodermal origin [2]. Various sources of MSCs include: bone marrow [3], adipose tissue [4], umbilical cord [5], dental pulp [6], and other tissues [7]. Due to their early developmental stage, wide differentiation capacity, various mechanisms of action, and relatively easy propagation ex vivo, MSCs, including UC-MSCs and ADSCs, represent cell populations with great potential for wide use in cellular-based therapies [8,9]. MSCs have been already successfully used in several clinical applications (see: clinicaltrial.gov database) and their beneficial effects were predominantly related to their paracrine activity [10]. However, MSC-based therapies may also be accompanied with some risk factors such as cell rejection, undesired immune response, toxicity [11], tumorigenic potential [12,13], prospective contamination with viruses [14], and problematic transport and storage of cells prior to transplantation [15,16,17]. To minimize potential risk factors related to the cellular nature of MSCs, some new cell-free approaches have been recently investigated [18,19].

Growing evidence has strongly indicated that most of MSC therapeutic effects were correlated with their paracrine activity. Among several released, soluble molecules, MSCs also produce small round vesicles surrounded by a phospholipid bilayer and carrying several bioactive components, called extracellular vesicles (EVs) [20]. 

EVs are released by majority of cells including prokaryotes [21], and multiple eukaryotic cells such as red blood cells [22], B and T cells [23,24], activated platelets [25], and several other mature mammalian cell types including stem cells. The main three subfractions of EVs may be mostly distinguished based on their diameter and include (i) larger apoptotic bodies (1 to 5 μm), (ii) smaller microvesicles also called ectosomes (100–1000 nm), and (iii) the smallest vesicles including predominantly exosomes (30–100 nm) [26]. EVs may be involved in several cell-to-cell communication strategies including juxtacrine signaling involving direct membrane interactions between EVs and target cell (e.g., via EV membrane ligands and target cells receptors) leading either to activation specific signaling pathway/s or exosome internalization by the target cell [27]. Importantly, it has been shown that EVs may transfer several bioactive molecules following the fusion with the target cell such as active transcription factors, enzymes, and various classes of RNAs including regulatory miRNAs [28]. Moreover, several analyses of EVs content have revealed that they may carry several types of membrane proteins including glycosylated adhesion molecules and receptors as well as lipids [29,30]. Importantly, it has been also discovered that EVs may even transfer whole organelles such as mitochondria, which sustained their functional properties in target cells following the intercellular transfer via EVs [31,32]. The two main classes of EVs, ectosomes and exosomes, represent the main fractions extensively studied in terms of their molecular content and functional properties. However, there are other types of membrane blebs including migrasomes and apoptotic bodies, which require further investigation. Migrasomes have been recently described as vesicles involved in cell migration arising during cell fibers retraction and may contain other cellular organelles such as smaller cytoplasmic vesicles [33]. Similarly, apoptotic bodies that are also considered today as cell-to-cell communication elements, may contain and transfer not only bioactive molecules, but also whole organelles including larger mitochondria and ribosomes [34,35]. The molecular content of EVs reflects characteristics of cells of their origin and eventually determines functionality of the vesicles in target cells. Therefore, differences between EVs released from different tissues are natural and in part predictable [28]. 

Importantly, growing evidence indicates that several populations of stem and progenitor cells including MSCs may also produce EVs, which may not only play a role in cell-to-cell communication, but also be utilized in regenerative approaches in tissue repair [36,37]. There are several potential advantages of EVs when compared to their parental cells such as: (i) inability to self-replicate implicating no risk of uncontrolled growth, (ii) limited potential to trigger immune system even in allo- and xeno-grafts, and (iii) easier transport and storage, which makes the potential EV-based, cellular-free treatment more optimal when compared to standard cell- based approaches [38,39].

Proteomic content of human umbilical cord MSC-derived extracellular vesicles (UC-MSC-EVs) consists of about five hundred (500) proteins originated from all subcellular localizations including the cytoplasm, membrane, nucleus, endoplasmic reticulum, mitochondrion, endosome, Golgi apparatus, and lysosome [40]. Almost one-fifth of them are involved in intracellular signaling cascade. Proteins regulating cell homeostasis, proteolysis, apoptosis, and protein localization constitute about ten percent of all proteins for each. A similar number of detected proteins in EVs are involved in cell adhesion, protein folding, and macromolecular assembling complex, and response to wound and organic substances [40]. Less than five percent of proteins are involved in inflammatory response, lipid and vesicle-mediated transport, translation, and cytoskeletal organization [40]. On the other hand, EVs from MSCs derived from adipose tissue (ADSC-EVs) have been shown to contain close to fifteen hundred (1500) proteins of similar intracellular origin [41]. The vast majority of ADSC-EV proteome, similar to UC-MSC-EVs, is involved in signaling, catalytic and regulatory activity. It has been shown that EVs from ADSCs may play a role in regulating more than two hundred signaling pathways, including phosphatidylinositol-3-kinase/protein kinase B (PI3K/Akt), Jak-STAT, and wingless-related integration site (Wnt) pathways, involved in guiding of cell fate, differentiation, and proliferation, as well as tissue regeneration [41].

## 2. Applications of UC-MSCs and ADSC EVs in Tissue Repair

Although, the field of potential use of EVs derived from stem cell populations including MSCs has been recently developed, several novel applications of such stem cell derivatives were indicated in tissue repair and EVs from both ADSCs and UC-MSCs are currently under increasing interest and investigation (Table 1). Selected representative approaches of MSC-EVs use in selected groups of conditions such as neurological, cardiovascular, liver, kidney, and skin diseases and injuries are summarized, along with the proposed mechanisms of EVs activity following their administration, which indicate potential future directions in this area.

### 2.1. Neurological Diseases

Millions of patients worldwide are annually diagnosed with disorders of the nervous system. Neurodegenerative conditions are of special interest due to their affect on cognitive functions. Alzheimer’s disease (AD) is an elaborated neurodegenerative condition, caused by β-amyloid peptide (Aβ) accumulation outside neurons and neurofibrillary tangles (NFT) consisting of hyperphosphorylated tau protein in cells [69]. Another neurodegenerative disorder is amyotrophic lateral sclerosis (ALS) resulting in general in impairment of human motor system and death predominantly because of respiratory failure. There are two types of this disorder: sporadic (90–95%) and familial (5–10%). The second one is caused by several gene mutations and the most abundant is mutation of SOD1 gene encoding: copper/zinc ion-binding superoxide dismutase [46,70]. Superoxide dismutase 1 (SOD1) is an important enzyme which exert antioxidant protection effect in cells [71]. When mutation occurs, SOD1 molecules aggregate and accumulate in motor neurons [72,73]. 

Although enormous effort is currently invested worldwide to limit the progression of such diseases in suffering patients, this field still requires novel approaches focused on limiting major mechanisms of these diseases’ development including inflammation and protein aggregation. Both UC-MSC- and ADSC-derived EVs may potentially represent new therapeutic agents interrupting the processes and as such limiting the diseases progression.

Ma et al. study of rat models of sciatic nerve transection injury showed that UC-MSC-EVs improved regeneration of nerve axons by inducing proliferation of Schwann cells and increasing myelination [42]. Importantly, the transplanted EVs modulated the profile of pro- and anti- inflammatory cytokines decreasing neuroinflammation and the neural tissue injury. There are two main cytokines identified to be involved in the process of the injury: pro-inflammatory interleukin 6 (IL-6) and IL-1β, whose levels decreased after treatment with UC-MSC-EVs, and anti-inflammatory IL-10, which concentration was increased [42]. 

Research study conducted by Ding and colleagues showed that UC-MSC-EVs could be beneficially used in Alzheimer’s disease treatment, upregulating the expression of neprilysin and insulin–degrading enzyme (IDE)—the two enzymes involved in β-amyloid peptide (Aβ) degradation [43]. The authors also observed that UC-MSCs-EVs regulated neuroinflammation by increasing levels of IL-10 and TGF-β, along with downregulation of TNFα and IL-1β [43]. Similar anti-inflammatory effect of UC-MSC-EVs, related to regulation of the same cytokine profile, has been confirmed by Shiue et al. in rat model of nerve injury-induced neuropathic pain [38] and by Sun et al. in murine models of spinal cord injury [44]. 

A study by Thomi and colleagues focused on perinatal brain injury [45], a condition occurring as a result of preterm birth [74], or neurodevelopment impairment [75]. The authors found that EVs derived from human Wharton’s jelly MSCs, exhibited anti-inflammatory effect on microglial cells treated in vitro as well as in rat model of perinatal brain injury in vivo by decreasing levels of IL-6, IL-1β and TNFα [45]. 

Thus, the immunomodulatory activity of UC-MSC-EVs following the transplantation into damaged neural tissues in both acute and chronic/neurodegenerative conditions represents their major mode of action, which may be accompanied with other regulatory capacity related to EVs’ molecular cargo. Similar anti-inflammatory effect has been postulated when bone marrow (BM) MSC-derived EVs were used in treatment of acute neural tissue injuries such as stroke and focal brain damage [76,77].

Interestingly, studies employing neuroblastoma N2a cell model in vitro, have shown that ADSC-EVs carry neprilysin—an important enzyme involved in Aβ peptide degradation in brain, which may be transferred by vesicles to these cells modulating the level of Aβ [46]. This finding strongly indicates that ADSC-EVs could be potentially useful in treatment of Alzheimer’s disease. According to the results of other study focusing on cultured primary brain cells from G93A mice carrying ALS phenotype, EVs from ADSC were capable to reduce level of harmful SOD1 aggregates in such neural cells and upregulate levels of phosphorylated cAMP response element-binding protein (p-CREB) and peroxisome proliferator-activated receptor gamma coactivator 1-alpha (PGC-1α)—the mitochondrial proteins, which are decreased during ALS development and progression [47]. Similarly, Bonafede and colleagues have shown in their study on ALS model in vitro, that ADSC-EVs provided neuroprotective effect and increased viability of motorneuron-like cell line NSC-3 subjected to cytotoxic hydrogen peroxide inducing cells apoptosis, suggesting antioxidative mechanism of EVs action [48]. 

Chen and colleagues, in their study, used exosomes derived from ADSCs in treatment of acute ischemic stroke (AIS) in rats in vivo [49]. They found that after ADSC-EVs treatment, brain infarct area was greatly decreased and neuronal function was partially restored. The expression of endothelial nitric oxide synthase (eNOS), vascular endothelial growth factor (VEGF), and chemokine receptor type 4 (CXCR4), which are molecules involved in angiogenesis and cell migration, were also increased. Additionally, no immune reaction or injury was recognized in major organs like the liver, brain, heart, kidney, and lungs, as potential side effect of the EVs administration. A study by Laso-García and colleagues in mice model of multiple sclerosis (MS), a chronic autoimmune disease affected central nervous system and causing demyelization of axons [78], showed that EVs derived from ADSCs enhanced motor function in mice, modulated neuroinflammation by decreasing levels of pro- and anti-inflammatory cytokines such as granulocyte-macrophage colony-stimulating factor (GM-CSF), IFNγ, TNFα, IL-1β, IL-2, IL-4, IL-5, IL-6, IL-12p70, IL-13, and IL-18, as well as reduced brain atrophy [50]. These results indicate pro-regenerative potential of EVs derived from ADSCs, which may be utilized not only for treatment of acute brain injuries, but potentially also neurodegenerative diseases. Interestingly, it has been shown that combined therapy with ADSCs and ADSC-EVs may be more effective in treatment of acute brain injuries, when compared to separate use of the cells or EVs [49], which also should considered when future strategies are optimized for therapies of neural system disorders.

### 2.2. Cardiovascular Diseases

Cardiovascular diseases (CVDs) have represented one of the major causes of motility and morbidity in western societies for several years. One of the most common acute cardiovascular disorder is acute myocardial infarction (AMI), caused by insufficient blood supply in heart tissue leading to massive cardiac cell necrosis and apoptosis resulting in progressing adverse myocardial remodeling and even death [79]. Additional damage to the ischemic myocardium may result from reperfusion, occurring when blood flow returns to heart tissue and causing tissue damage by oxidative stress and inflammation [80]. Although, several interventional and pharmacological approaches have been development for treatment of patients with acute and chronic myocardial injury, novel approaches focused not only on heart tissue restoration, but also on cytoprotection of myocardium in first stages of injury and accompanied with stimulation of endogenous mechanisms of repair, have to be further developed to provide optimal multidirectional treatment for such patients. Stem cell-based therapies including UC-MSCs and ADSCs and their derivatives such as EVs, represent such a new branch of complementary treatment for patients with CVDs.

Our previous studies have shown that extracellular vesicles from UC-MSCs provided cytoprotective effect on human primary heart cells in vitro as well as enhanced their differentiation capacity [51]. Other study employing rat model of AMI suggests that UC-MSC-EVs provide pro-differentiation effect on fibroblast, promoting myofibroblastic phenotype required in initial stages of the myocardial healing and decrease inflammatory response in infarct zone [52]. In an in vitro model employing rat cardiac myocytes (H9C2 cells), it has been shown that exosomes derived from UC-MSCs mediate anti-apoptotic effect increasing B-cell lymphoma 2 (Bcl-2) level and decreasing Bcl-2 associated X (Bax) protein in these cells [53]. Bcl-2 and Bax are proteins involved in regulation of cell apoptosis and survival [81] and the balance of their expression may be crucial in initial stages of myocardial injury, during a “fight” for part of post-AMI “hibernated myocardium” to survive ischemia/ reperfusion injury. 

UC-MSC-EVs have also been indicated to provide pro-proliferative effect in cells residing in border zone area of infarcted myocardium, as shown in rat models of AMI in vivo by increased level of Ki67 indicating cardiac cell proliferation [53]. A study conducted by Wang et al. revealed that EVs from UC-MSCs enhance Smad7 expression via inhibiting its impeditory miR-125b-5p, which eventually resulted in decline in myocardial cells apoptosis and injury [54]. Interestingly, study from Liu and colleagues suggests that UC-MSC-EVs may also regulate autophagy in cardiac cells by activating PI3K/AKT/mTOR signaling pathway leading to inhibition of heart cells apoptosis in rat model [55].

Similarly, it has been shown by Cui et al. that EVs derived from ADSCs may protect cardiac cells from ischemia/reperfusion (I/R) injury by activating Wnt/β-catenin signaling pathway [56], which was suggested to be involved in myocardial cells survival and regulation of apoptosis [82]. Interesting strategies may represent genetic modifications of parental cells such as MSCs to produce EVs enriched in selected regulatory miRNAs. A study by Luo and colleagues representing this trend has shown that areas of post-AMI infarcted myocardium and fibrosis, may be decreased following the treatment with miR-126 enriched ADSC-EVs [57]. Interestingly, we have also observed in our studies, significantly increased recovery of blood perfusion in ischemic muscles in murine model of ischemic limb injury in vivo, when UC-MSC-EVs overexpressing miR-126 were administrated intramuscularly, suggesting pro-angiogenic effect of vesicles [83]. 

Thus, the main mechanisms of MSC-derived EV activity in ischemic tissues include not only anti-inflammatory effects, but also cytoprotective, anti-apoptotic, and proangiogenic effects, increasing cell viability and tissue perfusion following EVs administration in vivo.

### 2.3. Liver Diseases

Several liver injuries also represent a relatively large problem in western societies, where EV applications may find their optimal niche for treatment. Liver fibrosis is an important adverse consequence of several acute liver injuries and chronic disorders and may occur as a result of hepatitis caused by viral infection, alcohol or drug abuse, bile flow impairment, or autoimmune reaction directed against hepatocytes [84]. Liver tissue fibrosis develops chronically with time, predominantly due to epithelial-mesenchymal transition (EMT) of hepatocytes, which transform to fibroblasts after e.g., activation signaling pathway induced by TGF-β1 [58,85]. On the other hand, acute liver failure may develop rapidly as a result of toxic drug effect or virus infection, even in individuals with no previous liver disorders. Although, it is a rare condition it may lead to death [86], if not treated immediately with possible multidirectional approaches.

A Study employing murine model of liver fibrosis in vivo, has indicated that transplantation of UC-MSC-EVs did not induce immune response, but decreased liver fibrosis by reducing TGF-β1 expression and reversion of EMT process [58]. Another study conducted by Yan et al. focusing on both in vivo (murine model of acute toxicant-induced liver injury) and in vitro (human cell lines: fetal hepatocyte (L02) and lung fibroblasts (HFL1)) models, has shown that glutathione peroxidase 1 (GPX1) carried by UC-MSC-EVs protected from liver failure caused by carbon tetrachloride (CCl_4_) administration [59], which suggested anti-oxidative effect of EVs. Furthermore, UC-MSC-EVs reduced oxidative stress and inflammation by down-regulation of pro-inflammatory cytokines: IL-1α, IL-6, TNF-α, and monocyte chemoattractant protein-1 (MCP-1) [59]. The authors have also indicated that anti-apoptotic effect of UC-MSC-EVs was related to their ability to (i) induce extracellular signal–regulated kinases (ERK)1/2 phosphorylation, (ii) activate Bcl2 expression (mechanism that protects cells from apoptosis [87,88]), and (iii) downregulation of caspases (Casp) 9 and 3 expression [59]- proteins involved in apoptosis caused by e.g., oxidative stress [89].

Similarly, ADSC- derived EVs may reduce fibrosis and inflammation following administration into injured liver tissue as shown by Qu et al. [60]. The results obtained in murine model of CCl_4_ (toxicant)- induced liver injury in vivo suggested that selected microRNAs transferred form ADSC-EVs to liver cells (e.g., miR-181-5p) may enhance autophagy and downregulate expression of proteins involved in fibrosis and inflammation such as collagen I, vimentin, alpha–smooth muscle actin (α-SMA), Bcl-2, TNFα, IL-6, IL-17 [60]. 

Thus, in addition to the direct anti-fibrotic effect, the major mechanisms of UC-MSC-EVs and ADSC-EVs activity in injured liver tissues are similar to the ones previously described in neurological and cardiovascular injuries, and include cytoprotective, anti-oxidative and anti-inflammatory capacity of EVs.

### 2.4. Kidney Diseases

Kidney diseases represent abundant group of disorders causing severe health consequences for patients, leading often in progressed stages even to death due to tissue toxification, if any donor kidney transplantation is impossible or fails. Ischemia/reperfusion injury is the most common cause of acute kidney failure [40], which develops rapidly (within days) and manifests in high serum creatinine level and decrease in urine production [90]. Acute kidney injury often progresses into chronic kidney disease (CKD) with time, which is manifested with impaired kidney function lasting more than 3 months [91]. Kidney failure may also be caused by more complex disorders like renal artery stenosis (RAS) and metabolic syndrome (MetS) [63], which complicate the primary diseases progression and manifestation. Because kidney failures are often associated with significant patient discomfort and dialysis dependency, scientists are currently questing for new supportive treatments, including stem cells and EVs applications.

A study conducted by Kilpinen et al. employing rat model of acute kidney injury following ischemia and reperfusion (I/R) in vivo has shown that EVs from UC-MSCs were able to suppress proliferation of T-cells and to induce their regulatory phenotype [40]. The authors also showed that UC-MSC-EVs administration decreased the level of the injury-related markers such as serum creatinine, serum urea, aspartate aminotransferase and gamma glutamyltransferase, which indicated partial recovery from functional renal failure [40]. Other studies focusing on renal epithelial NRK-52E cells in vitro and rat model of acute kidney injury (AKI) in vivo, have indicated that UC-MSC-EVs reduced kidney damage caused by cisplatin (potentially toxic anticancer drug) via their anti-apoptotic and anti-oxidative activity [61]. They found that administration of the EVs increased Bcl-2 expression, along with decreased expression of Bax, which protected kidney cells from apoptosis. Additionally, the level of glutathione (GSH) was increased, along with decreased levels of malondialdehyde (MDA) and oxidative stress product such as 8-hydroxy-2′-deoxyguanosine (8-OHdG) after treatment with UC-MSC-EVs, indicating attenuation of oxidative stress [61]. 

Interestingly, the first clinical study in humans recently conducted with forty patients with chronic kidney disease (CKD) in stage III and IV showed that UC-MSC-EVs were safe in clinical use and improved kidney function in the group treated with EVs, when compared to placebo – control group [62]. The authors of the trial have found that UC-MSC-EVs administration increased the level of estimate glomerular filtration rate (eGFR), as well as plasma levels of anti-inflammatory IL-10 and TGF-β [62]. Furthermore, levels of blood urea, serum creatinine, and pro-inflammatory TNF-α were decreased. These important results from the first clinical trial in humans suggest that kidneys functions in patient with CKD may be partially restored following the treatment with UC-MSC-EVs [62], which opens potential new perspectives for such patients with chronic/ late stage of kidney disorders, if the results are confirmed in next, large scale, controlled clinical studies.

Similarly, Eirin et al. found that ADSC-EVs had positive impact on functional recovery of kidneys in swine model of metabolic syndrome (MetS) and renal artery stenosis (RAS) in vivo [63]. EVs application caused decline in the levels of pro-inflammatory cytokines such as MCP-1, TNF-α, IL-6, IL-1β and normalization of the levels of IL-4, IL-10 indicating an anti-inflammatory effect. This was accompanied by a decrease in the level of serum creatinine, which is a marker of renal injury [63]. Another study conducted by the same research group employing similar swine model in vivo revealed that EVs from ADSCs improved kidneys functions via (i) promoting vascular maturation, (ii) increase of expression of pro-angiogenic factor such as VEGF and also other pro-regenerative factors (neurogenic locus notch homolog protein 1 [Notch1], delta-like protein 4 [DLL4])) and (iii) restoring microcirculation [64]. 

Another study by Lin et al. employing rat model of AKI in vivo indicated that ADSC-EVs attenuated inflammation in kidney tissues by decreasing levels of TNF-α, nuclear factor kappa-light-chain-enhancer of activated B cells (NF-κB), and IL-1β [65]. However, this study also showed that combined administration of ADSCs and ADSC-EVs may be more effective in acute kidney injury treatment, when compared to the single use of cells or EVs separately [65]. This is an important finding for further optimization of the stem cell- and EV- based therapeutic strategies in patients with kidney disorders and injuries.

To sum up, similar to the previously discussed applications of UC-MSC- and ADSC- derived EVs in other disorders, the main mechanism of EVs’ action in kidney tissues include anti-inflammatory activity that may be accompanied with anti-oxidative and pro-angiogenic effects mediated by the vesicles and their molecular cargo. Interestingly, similar mechanisms have been described when BM-MSC-derived EVs were administrated in animal models of acute and chronic kidney injuries in vivo [92].

### 2.5. Skin Diseases

Several minor skin injuries including wounds and burns often plague people in daily life. However, in some adverse circumstances skin injuries may be extensive including large burns covering significant body surface, which makes them complex and difficult to treat with standard procedures. Wounds and burns may be often accompanied with pain, infections and result in scars formation. Therefore, new approaches enhancing healing and favoring scarless new skin tissue formation, need to be optimized for such patients along with standard procedures relying on skin transplantation and pharmacological treatment. Novel applications of MSC-derived EVs are currently being considered as a potential promising tool for such therapies for skin injuries.

According to research by Zhang et al., EVs derived from UC-MSCs may enhance proliferation and survival of human keratinocytes and dermal fibroblasts in vitro as well as promote skin regeneration in rat model of skin burn in vivo by combined activation of AKT and Wnt4/β-catenin signaling pathways [66] involved in skin cell survival, proliferation and wound healing [93,94]. It has been also shown by Kim et al. that UC-MSC-EVs may functionally stimulate normal human dermal fibroblasts (HDFs) in vitro by enhancing production of extracellular matrix (ECM) components such as collagen, fibronectin, and elastin [95]. Additionally, the authors have shown lower production of matrix metalloproteinase-1 (MMP1) in fibroblasts treated with UC-MSC-EVs that may be potentially utilized for improvement of wound healing, if required due to injury [95].

Similarly, it has been reported that in ex vivo rat models of skin flap exposition to ischemia/reperfusion injury, EVs derived from ADSCs improved the skin pieces survival by enhancing neovascularization inside the tissue [67]. Moreover, the authors found that preconditioning ADSCs with low doses of H_2_O_2_ escalated pro-regenerative effects mediated by their EVs [67]. 

Another study employing mice model of full-thickness incision of skin flap in vivo indicated that ADSC-EVs may have a beneficial impact on skin regeneration through increasing collagens and MMP-1 production as well as by activation of PI3K/Akt signaling pathway [68]. Moreover, similarly to the previously described study, ADSC-EVs had a beneficial influence on neovascularization during wound healing in vivo [68].

## 3. Conclusion and Future Prospective

Our synthetic review points to the most important mechanisms of extracellular vesicles activity, which has been identified following their administration in several neurological, cardiovascular, liver, kidney, and skin injuries. Moreover, there is currently growing evidence supporting important role of extracellular vesicles derived from ADSCs and UC-MSCs in other conditions including in bone regeneration [96,97,98] and cancers [99,100,101], indicating similar mechanisms to be involved (summarized in Figure 1).

MSC-derived EVs may play their role by following mechanisms of action after transplantation: (i) regulation of inflammatory response (increase in level of anti-inflammatory cytokines and decrease in pro-inflammatory ones), (ii) increasing endogenous cells proliferation, differentiation and viability in place of injury (e.g., by autophagy regulation), (iii) reducing oxidative stress, and iv) enhancing angiogenesis in place of injury, which together may contribute to tissue repair and prevent organs and tissues from severe anatomical and functional damages (Figure 1).

Considering the fact that MSC-derived EVs may modulate various processes that may have a dual role in tissue structure remodeling, their potential future therapeutic applications need to be optimized in terms of the dose, time window after injure and most importantly the place of transplantation. For instance, stimulation of fibroblast proliferation and increase in deposition of ECM proteins described in skin after MSC-EVs injection, may not be beneficial effect in other tissue types such as infarcted myocardium or injured brain or liver tissues, where may lead to adverse remodeling, fibrosis, and scar formation. Thus, controlled and optimized applications of MSC-EVs in tissue repair are still challenging and require further investigations. 

Extracellular vesicle-based therapies may represent promising future therapeutic approaches in regenerative medicine due to their bioactive cargo and potential wide range of applications in several disorders and injuries, which have already been indicated by several investigators based on in vitro and in vivo studies. Foregoing research publications have given reliable indications that EVs produced by MSCs of various origin may exhibit similar manner of paracrine activity to their parental cells. However, several challenges and potential risk factors should be still considered when thinking about future applications of MSC-EVs in tissue repair, including challenging harvesting and purification of EVs from various types of specimens (culture media, serum, urine, milk, and other potential sources). Additionally, due to the still limited number of clinical studies employing EV- based therapies (see: clinicaltrial.gov database), the safety, immunogenicity or optimal doses of EVs are still unknown. Other challenges include scaling up of EV production, optimizing, and set up of GMP procedures and their safety analysis. Importantly, further studies with side-by-side comparison of various EV specimens (including UC-MSC-EVs vs. ADSC-EVs) in various tissue injury models, are still required to indicate most potent EVs specimens for particular tissue repair. Despite the current challenges, we believe that EV specimens may be successfully used in the future in selected approaches of tissue repair and may partially replace whole cell- based applications, when such EV- based therapeutic products are optimized and well defined in their bioactive cargo and mechanisms of action.

## Figures and Tables

**Figure 1 ijms-21-00799-f001:**
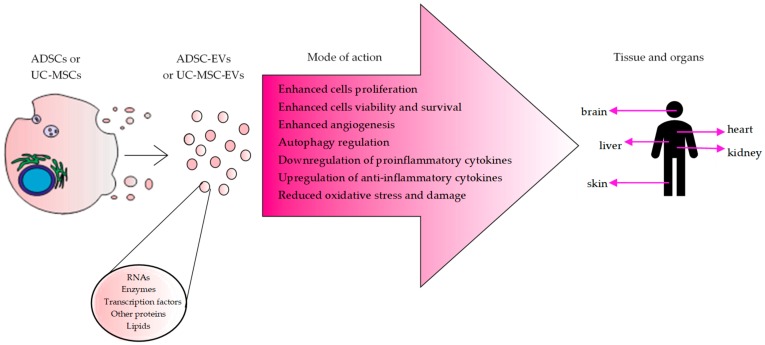
Extracellular vesicles mode of action in various tissues.

**Table 1 ijms-21-00799-t001:** Possible applications of adipose tissue (ADSCs) and umbilical cord (UC-MSCs) extracellular vesicles in tissue repair.

TYPE OF DISEASE	SOURCE OF EVS	INJURY/ DISEASE	MODEL OF DISEASE	MECHANISM OF EV ACTION	REFERENCE
Neurologic-al	UC-MSCs	Sciatic nerve transection injury	in vivo (rat)	Induction Schwann cell proliferation, anti-inflammatory activity (downregulation of IL-6, IL-1β, upregulation of IL-10)	Ma et al. 2019 [42]
		Alzheimer’s disease (AD)	in vivo (mice)	Enhanced Aβ degradation, anti-inflammatory activity (downregulation of TNFα, IL-1β, upregulation of IL-10 and TGF-β)	Ding et al. 2018 [43]
		Spinal nerve ligation (SNL) pain model	in vivo (rat)	Suppression of neuroinflammation (downregulation of IL-1β and TNFα)	Shiue et al. 2019 [38]
		Spinal cord injury	in vivo (mice)	Anti-inflammatory activity (downregulation of TNFα, IL-6, IFNγ1, G-CSF2, MCP-1, MIP-1α3, upregulation of IL-4, IL-10)	Sun et al. 2018 [44]
		Perinatal brain injury	in vivo (rat), in vitro (BV-2 micro-glial cells)	Suppression of neuroinflammation (downregulation of IL-6, IL-1β and TNFα)	Thomi et al. 2019 [45]
	ADSCs	Alzheimer’s disease (AD)	in vitro (neuro-blastoma N2a cells)	Enhanced Aβ degradation by neprilysin present on EVs’ surface	Katsuda et al. 2013 [46]
		Amyo-trophic lateral sclerosis (ALS)	in vitro (G93A mice primary neuronal stem cells)	SOD1 aggregates degradation, upregulation of p-CREB and PGC-1a (mitochondrial protection)	Lee et al. 2016 [47]
		Amyo-trophic lateral sclerosis (ALS)	in vitro (moto-neuron-like NSC-34 cells)	Increased cell viability (cytoprotection from oxidative damage)	Bonafede et al. 2016 [48]
		Acute ischemic stroke	In vivo (rat)	Decreased brain infarct area, increased levels of eNOS, VEGF, CXCR4	Chen et al. 2016 [49]
		Multiple sclerosis (MS)	In vivo (mice)	Decreased levels of pro- and anti-inflammatory cytokines and brain atrophy; improved global animal motor activity	Laso-García et al. 2018 [50]
Cardiovasc-ular	UC-MSCs	Myocardial ischemia	in vitro (human primary heart cells)	Increased cardiac cell proliferation, differentiation, and survival in cytotoxic conditions	Bobis-Wozowicz et al. 2017 [51]
		Acute myocardial infarction (AMI)	in vivo (rat)	Promotion of fibroblast - to -myofibroblast differentiation, cardiomyocyte cytoprotection	Shi et al. 2019 [52]
		Acute myocardial infarction (AMI)	in vivo (rat) in vitro (cardio – myoblast cell line H9C2)	Improving cardiac systolic function due to anti-apoptotic and proangiogenic effects (related to e.g., Bcl-2 family expression)	Zhao et al. 2015 [53]
		Acute myocardial infarction (AMI)	in vivo (rat)	Decrease in myocardial cells apoptosis and injury (by, e.g., reducing level of miR-125b)	Wang et al. 2018 [54]
		Myocardial ischemia	in vitro (rat cardio – myoblast cell line)	Anti-apoptotic effect via regulating autophagy by PI3K/AKT/mTOR signaling pathway activation	Liu et al. 2019 [55]
	ADSCs	Myocardial ischemia	in vivo (rat) in vitro (rat, cardio – myoblast cell line H9C2)	Increasing cardiac cell survival by Wnt/b-catenin signaling pathway activation and regulation of Bcl-2/Bax gene expression	Cui et al. 2017 [56]
		Acute myocardial infarction (AMI)	in vivo (rat) in vitro (cardio – myoblast cell line H9C2)	Tissue fibrosis inhibition, by miR-126 transfer	Luo et al. 2017 [57]
Liver	UC-MSCs	Toxicant- induced liver injury	in vivo (mice)	Decrease in liver fibrosis by downregulation of TGF-β1 expression, inhibiting EMT, and hepatocyte cytoprotection	Li et al. 2013 [58]
		Toxicant- induced liver injury	in vivo (mice) in vitro (mice, HFL1, L02 cell lines)	Alleviate liver failure via antioxidant and anti-apoptotic effects (e.g., by GPX1 transfer, downregulation of IL-1α, IL-6, TNF-α)	Yan et al. 2017 [59]
	ADSCs	Toxicant- induced liver injury	in vivo (mice) in vitro (mice, hepatic stellate cells, HST-T6)	Decrease in liver fibrosis by downregulation of collagen I, vimentin, α-SMA, TNFα, IL-6, IL-17 and fibronectin, and autophagy activation (due to, e.g., mir-181-5p transfer)	Qu et al. 2017 [60]
Kidney	UC-MSCs	Acute kidney injury (AKI)	in vivo (rat)	Cytoprotective and anti-inflammatory activity via suppression of T-cells proliferation	Kilpinen et al. 2013 [40]
		Acute kidney injury (AKI)	in vivo (rat) in vitro (NRK-52E cells)	Cytoprotective, anti-oxidative, and anti-apoptotic effects (e.g., by upregulation of Bcl-2, GSH and downregulation of Bax, MDA, 8-OHdG), promoting epithelial cell proliferation	Zhou et al. 2013 [61]
		Chronic kidney disease (CKD)	In vivo (humans, clinical study, first in humans)	Enhanced overall kidney function via decreasing immune response (e.g., by upregulation of IL-10, TGF-β, and down- regulation of TNF-α), no side effects detected	Nassar et al. 2016 [62]
	ADSCs	Metabolic syndrome (MetS) and renal artery stenosis (RAS)	in vivo (swine)	Renoprotective effects via anti-inflammatory activity (e.g., by upregulation of IL-4, IL-10 and downregulation of MCP-1, TNF-α, IL-6, IL-1β)	Eirin et al. 2017 [63]
		Metabolic syndrome (MetS) and renovascu-lar disease (RVD)	in vivo (swine)	Restoring hemodynamics and renal function via anti-apoptotic, antioxidative, and proangiogenic effects (e.g., by upregulation of VEGF, Notch1, DLL4, increased vascular maturation, microcirculation)	Eirin et al. 2018 [64]
		Acute kidney injury (AKI)	in vivo (rat)	Renoprotective effect via anti-inflammatory activity (e.g., by downregulation of TNF-α,IL-1β)	Lin et al. 2016 [65]
Skin	UC-MSCs	Skin burn	in vivo (rat) in vitro (HaCAT, HFL1 and DFL cells)	Accelerated re-epithelialization due to increase in skin cell survival and proliferation (via activation of AKT and Wnt4/β-catenin pathways, respectively)	Zhang et al. 2015 [66]
	ADSCs	Skin flap ischemia/ reperfusion injury	in vivo (rat)	Recovery of skin flap following I/R via increasing cell survival, neovascularization, and decreasing apoptosis and inflammation in the tissue	Bai et al. 2018 [67]
		Skin Wound (Full-thickness incision of skin flap)	in vivo (mice) in vitro (human dermal fibro-blasts, HDFs)	Accelerating full- thickness wound healing via increase in fibroblast proliferation, migration and collagen deposition (e.g., by upregulation of collagens, MMP-1 levels, activation of PI3K/Akt signaling pathway)	Zhang et al. 2018 [68]

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
