# Peer review of "Perspectives for Future Use of Extracellular Vesicles from Umbilical Cord- and Adipose Tissue-Derived Mesenchymal Stem/Stromal Cells in Regenerative Therapies—Synthetic Review"

_ijms, 2020, doi:10.3390/ijms21030799_

Round 1
Reviewer 1 Report
Line 48. The authors should pay attention to experimental works instead/in addition to reviews (#11,12).
There are some typos (for example, line 248 - UC-MAC-EV). Line 51. “was correlated” should be “were correlated”.
Line 50. Mention an appropriate work, for example (doi: 10.1007/s12668-016-0348-0).
Line 67. The authors should mention that EVs contain organelles (mitochondria, ribosomes and proteasomes).
Section 2.1. Neurological Diseases. The authors did not pay attention to treatment of multiple sclerosis with MSCs derived EVs which demonstrate significant immunomodulatory activity.
Conclusion Section. The authors should discuss the multifaceted role of EVs in fibroblast proliferation and collagen deposition in skin, myocardial damage and liver fibrosis (Table 1). EVs posses stimulatory activity in first case (Table 1, Lines 188, 324-326) and inhibitory in second (Table 1). This will improve the impact of this review.
Describing of limitations, shortcomings and potential problems associated with the use of EVs for disease treatment will improve the manuscript and bring a larger audience in this field.
Author Response
RESPONSE TO THE COMMENTS OF REVIEWER 1
We would like to thank the Reviewer for his/her critique, which has helped us to improve the quality of our manuscript. We would also like to thank the Reviewer for his/her kind remarks regarding the significance of concepts presented in this manuscript.
Minor Comments:
Line 48. The authors should pay attention to experimental works instead/in addition to reviews (#11,12).
We thank the Reviewer for this suggestion. We agree that additional experimental papers should be cited. Thus, we included additional experimental papers, which describes potential risk factors connected with MSCs utilization as therapy (page 2, lines: 47-49). Additional references were also added in reference section; in red.
There are some typos (for example, line 248 - UC-MAC-EV). Line 51. “was correlated” should be “were correlated”.
We apologize for those mistakes. We changed indicated typos (page 2, lines: 51; page 9, line: 265) and also additional three: there was “in vitro” and we changed for “in vivo” (in Table 1, in section with neurological diseases (page 4); a typo in neprilysin (page 7, line: 140); and “possible” changed to “impossible” (page 9, line: 273). All changes are hallmarked in red.
Line 50. Mention an appropriate work, for example (doi: 10.1007/s12668-016-0348-0).
Following the Reviewer’s suggestion, we included into the manuscript two additional citations, one proposed by Reviewer and the additional one (page 2, line: 50). Additional references were also added in reference section; in red.
Line 67. The authors should mention that EVs contain organelles (mitochondria, ribosomes and proteasomes).
We thank the Reviewer for this remark. This important aspect was also considered. Thus, we added more information related to this phenomenon (page 2, lines: 66-77) and also additional references were included in reference section. All changes are in red.
Section 2.1. Neurological Diseases. The authors did not pay attention to treatment of multiple sclerosis with MSCs derived EVs which demonstrate significant immunomodulatory activity.
We thank the Reviewer for this important comment. We included into the text the information about example of MSC-EV use in the treatment of multiple sclerosis in mice model (page 7, lines: 177-183). Also, additional row was added into the Table 1, in section of neurological diseases (page 4) and two additional abbreviations was added in appropriate section (page 13); new references in reference section was included. All changes are in red.
Conclusion Section. The authors should discuss the multifaceted role of EVs in fibroblast proliferation and collagen deposition in skin, myocardial damage and liver fibrosis (Table 1). EVs posses stimulatory activity in first case (Table 1, Lines 188, 324-326) and inhibitory in second (Table 1). This will improve the impact of this review.
We thank the Reviewer for this very important comment. We absolutely agree that this aspect should be discussed. Thus, we included the additional part of text briefly considering this aspect in Conclusion and Future Prospective section (page 12, line: 373-380). We hope it meets the Reviewer’s requirements. All changes are in red.
Describing of limitations, shortcomings and potential problems associated with the use of EVs for disease treatment will improve the manuscript and bring a larger audience in this field.
We thank the Reviewer for this suggestion. We agree that adding information about limitations, shortcomings and potential problems may be an important aspect for readers interested in the a article. Thus, we added a comment in Conclusion and Future Prospective section (page 12, lines: 386-395). All changes are in red.
We hope that all our explanations provided above as well as the changes in our manuscript to improve its quality, would meet the Reviewer’s requirements.
Reviewer 2 Report
The manuscript "Perspectives for Future Use of Extracellular Vesicles from Umbilical Cord- and Adipose Tissue- Derived Mesenchymal Stem/ Stromal Cells in Regenerative Therapies – Synthetic Review" presented by J. Lelek and E. K. Zuba-Surma was aimed to review the current research conducted in vitro and in vivo by employing animal models of some neurological, cardiovascular, liver, kidney and skin diseases and within initial clinical trials. The review provides an analysis of outcomes for studies of pro-regenerative capacity of the MSC- extracellular vesicles in various models of diseases mediated by several mechanisms.
The article may be interesting to a wide circle of researchers and physicians. The information provided in this manuscript may be useful for further research. The manuscript gives an overview of the latest findings of the field. The references were used properly. Therefore, I recommend the manuscript for publishing in this journal. However, there are some minor points to be addressed.
This review discusses the effects of extracellular vesicles produced by two types of mesenchymal cells - UC-MSC and AD-MSC. The results of studies of these effects presented in Table 1 are analyzed in sections devoted to various diseases. However, the authors do not consider the similarities and differences in the effects of exposure to extracellular vesicles from these two cell populations. In my opinion, these properties should be emphasized in the conclusion paragraph of each section or in the final conclusion section. Figure 1 is very simplified, whereas it needs to be more detailed. It is necessary to indicate the cellular sources of MSCs, as well as the examined tissues and organs that respond to extracellular vesicular factors. It would be useful to list the factors contained in the vesicles that affect these cellular processes.Author Response
RESPONSE TO THE COMMENTS OF REVIEWER 2
We would like to thank the Reviewer for his/her kind opinion and valuable critics, which have helped us to improve the quality of our manuscript. We would also like to thank the Reviewer for his/her kind remarks regarding the significance of concepts presented in this manuscript.
Minor Comments:
This review discusses the effects of extracellular vesicles produced by two types of mesenchymal cells - UC-MSC and AD-MSC. The results of studies of these effects presented in Table 1 are analyzed in sections devoted to various diseases. However, the authors do not consider the similarities and differences in the effects of exposure to extracellular vesicles from these two cell populations. In my opinion, these properties should be emphasized in the conclusion paragraph of each section or in the final conclusion section.
We thank the Reviewer for this significant comment. We agree that discussion this aspect would be beneficial and improve manuscript. However, comparative side-by-side studies are very few and more research is still required. We indicated this problem and the need for more future comparative studies in Conclusion and Future Prospective section (page 12, lines: 392-395). We hope this additional information would meet the Reviewer’s requirements.
Figure 1 is very simplified, whereas it needs to be more detailed. It is necessary to indicate the cellular sources of MSCs, as well as the examined tissues and organs that respond to extracellular vesicular factors. It would be useful to list the factors contained in the vesicles that affect these cellular processes.
We thank the Reviewer for this important comment. According to the Reviewer’s suggestion, we added a few details into Figure 1 (page 11, line: 370) to make it more readable and informative. New Figure 1 is included into manuscript.
We hope that all our explanations provided above as well as the changes in our manuscript to improve its quality, would meet the Reviewer’s requirements.